# BRST invariant formulation of the Bell-CHSH inequality in gauge field theories

D. Dudal,[1, *] P. De Fabritiis,[2, †] M. S. Guimaraes,[3, ‡] G. Peruzzo,[4, §] and S. P. Sorella[3, ¶]

[1]KU Leuven Campus Kortrijk–Kulak, Department of Physics,
Etienne Sabbelaan 53 bus 7657, 8500 Kortrijk, Belgium and Ghent University,
Department of Physics and Astronomy, Krijgslaan 281-S9, 9000 Gent, Belgium
[2]CBPF – Centro Brasileiro de Pesquisas Físicas,
Rua Dr. Xavier Sigaud 150, 22290-180, Rio de Janeiro, Brazil
[3]UERJ – Universidade do Estado do Rio de Janeiro,
Instituto de Física – Departamento de Física Teórica – Rua São
Francisco Xavier 524, 20550-013, Maracanã, Rio de Janeiro, Brazil
[4]UFF – Instituto de Física, Universidade Federal Fluminense,
Campus da Praia Vermelha, Av. Litorânea s/n, 24210-346, Niterói, RJ, Brazil

A study of the Bell-CHSH inequality in gauge field theories is presented. By using the Kugo-Ojima analysis of the BRST charge cohomology in Fock space, the Bell-CHSH inequality is formulated in a manifestly BRST invariant way. The examples of the free four-dimensional Maxwell theory and the Abelian Higgs model are scrutinized. The inequality is probed by using BRST invariant squeezed states, allowing for large Bell-CHSH inequality violations, close to Tsirelson's bound. An illustrative comparison with the entangled state of two 1/2 spin particles in Quantum Mechanics is provided.

## I. INTRODUCTION

The Bell-Clauser-Horne-Shimony-Holt inequality [1–3] is a cornerstone of Quantum Mechanics. Its violation signals the existence of strong correlations between the components of a composite system which cannot be accounted for by local realistic theories. This inequality plays a pivotal role in many aspects of modern Quantum Mechanics such as quantum computation, teleportation and quantum cryptography. In its traditional form, the Bell-CHSH combination reads [1–3]

$$\langle\psi|\mathcal{C}_{CHSH}|\psi\rangle = \langle\psi|(A_1 + A_2)B_1 + (A_1 - A_2)B_2|\psi\rangle, \quad (1)$$

where $|\psi\rangle$ is the quantum state of the system and $(A_i, B_k)$, with $i, k = 1, 2$, are four bounded Hermitian operators fulfilling the strong requirements [4, 5]

$$A_i^2 = 1, \qquad B_k^2 = 1, \qquad [A_i, B_k] = 0. \quad (2)$$

One speaks of a Bell-CHSH inequality violation whenever

$$|\langle\psi|\mathcal{C}_{CHSH}|\psi\rangle| > 2. \quad (3)$$

The value $2\sqrt{2}$ is the maximum allowed violation, being known as Tsirelson's bound [4]. In the last decades, the experimental evidences of Bell-CHSH inequality violations have reached a very high degree of sophistication, as one can see in Refs. [6–13], confirming the predictions of Quantum Mechanics. Generalizations of the inequality when the involved operators do not necessarily have ±1 eigenvalues can be found in [14].

Since the end of the eighties, the Bell-CHSH inequality has been investigated also from the relativistic Quantum Field Theory (QFT) point of view [15–19]. In particular, in the pioneering articles [15–19], the authors have been able to show that even free quantum field theories might lead to strong violations of the Bell-CHSH inequality. It should be pointed out that, although this subject is still barely addressed in the literature when we compare it with the developments made in the context of Quantum Mechanics, it has been receiving some contributions recently, cf. Refs. [20–22].

It is worth underlining that, besides the pure theoretical aspects, Bell-CHSH inequality violations are receiving a recent attention from both phenomenological and experimental sides. In fact, many experimental tests in the high-energy context were proposed recently, for instance, in systems with top-quarks [23–27], $\Lambda$ baryons [28], $\tau$ leptons and photons [29]. There are also proposals for testing experimentally the Bell-CHSH inequality in $e^+e^-$ collisions [30], neutral meson physics [31], neutrino oscillations [32], charmonium [33, 34] and positronium [35] decays, and in the Higgs boson decay into $W$ bosons [36–38], see also [39–41]. Therefore, there is a huge interest in investigating the Bell-CHSH inequality in the Particle Physics context, allowing us to discuss this subject at very high energy scales. Thus, this offer us the possibility of investigating entanglement in a regime never explored before, where the appropriate physical framework is deeply in the realm of QFT.

The aim of the present work is to pursue the study of the Bell-CHSH inequality in relativistic Quantum Field Theory. More precisely, we shall focus on the analysis of gauge field theories within the framework of BRST quantization which, as shown by Kugo-Ojima [42–45], provides a powerful tool in order to identify the invariant subspace corresponding to the positive norm physical states. The whole construction relies on the analysis of the BRST charge cohomology in Fock space [46].

* david.dudal@kuleuven.be
† pdf321@cbpf.br
‡ msguimaraes@uerj.br
§ gperuzzofisica@gmail.com
¶ silvio.sorella@gmail.com

The main result of the current investigation is the formulation of the Bell-CHSH inequality in a BRST invariant way, a feature which can be considered as a basic requirement for a consistent formulation of the Bell-CHSH inequality in gauge theories. This statement can be formalized through the following steps:

- a characterization of the BRST invariant states $|\psi\rangle$ entering the Bell-CHSH inequality (1) as nontrivial elements of the cohomology [46] of the BRST charge $Q_{BRST}$, namely

$$\mathcal{H}_{phys} = \{|\psi\rangle\; ;\; Q_{BRST}|\psi\rangle = 0,\; |\psi\rangle \neq Q_{BRST}|\cdot\rangle\}. \quad (4)$$

- a careful construction of the four bounded Hermitian operators $(A_i, B_k)$, $i, k = 1, 2$, which, in addition of requirements (2), have now to be BRST invariant as well, $i.e.$,

$$[Q_{BRST}, A_i] = 0, \qquad [Q_{BRST}, B_k] = 0. \quad (5)$$

We have tacitly assumed the operators $(A_i, B_k)$ to be bosonic, otherwise the anti-commutator would appear in Eq. (5).

As one can easily figure out, the combination of the conditions (4) and (5) results in a BRST invariant formulation of the Bell-CHSH inequality.

As gauge field theories, we shall discuss the massless Maxwell theory and the massive $U(1)$ Higgs model quantized in covariant gauges, in $1 + 3$ spacetime dimensions. In both cases we shall outline the construction of the physical subspaces as well as of the BRST invariant operators $(A_i, B_k)$ entering the inequality (1).

Concerning the state $|\psi\rangle$, we shall consider the example of BRST invariant squeezed states built with the BRST invariant ladder operators corresponding to the physical (transverse) polarizations of the Maxwell field and of the massive vector boson of the Higgs model. This kind of state will lead to a violation of the Bell-CHSH inequality close to saturating the Tsirelson's bound. As a motivation behind the choice of such states, we remark that photon squeezed states are known to be highly entangled, giving rise to applications in many fields [47], see also [48] for their recent use in gravitational wave detectors.

This article is organized as follows. In Section II we work out a toy model inspired by the Fujikawa model [49], consisting of a BRST quartet of massive fields in $1 + 1$-dimensional Minkowski spacetime, $i.e.$, a pair of BRST doublets [46]. As a consequence, the only nontrivial BRST invariant state turns out to be the vacuum state $|0\rangle_M$. Despite of its simplicity, the model has the merit of illustrating in a clear and pedagogical way the whole Kugo-Ojima construction. Section III is devoted to the analysis of the massless Maxwell field in $1 + 3$-dimensional Minkowski spacetime. We survey the Kugo-Ojima setup and identify the physical invariant subspace. We proceed then with the introduction of the squeezed states, of the BRST invariant operators $(A_i, B_k)$, and

with the analysis of the Bell-CHSH inequality violation. Section IV addresses the comparison with the entangled two spin 1/2 states in Quantum Mechanics. In Section V, we discuss the $U(1)$ Higgs model, generalizing the construction done in the case of the massless Maxwell field to the case of massive vector bosons. Section VI contains our conclusion and future perspectives.

## II. WARMING UP: BRST COHOMOLOGY AND KUGO-OJIMA QUARTET MECHANISM

In this section, we review a toy model based on [49] in order to highlight some important features related to the BRST symmetry [50, 51], setting the stage for the discussion to come. Readers already familiar with BRST quantization and the Kugo-Ojima construction can jump to Sect. III.

Defining $g_{\mu\nu} = \text{diag}(+1, -1)$ as the Minkowski metric in $1 + 1$ dimensions, we shall consider the massive model specified by the following action

$$S = \int d^2x \left[ \partial_\mu \varphi \partial^\mu \eta - m^2 \varphi \eta - \partial_\mu \bar{c} \partial^\mu c + m^2 \bar{c} c \right], \quad (6)$$

where $\varphi$ and $\eta$ are real scalars, $c$ and $\bar{c}$ are anti-commuting ghost fields and $m$ is a parameter with dimension of mass. This model is invariant under BRST transformations, defined by the following action of the nilpotent BRST operator $s$ on the fields:

$$\begin{aligned} s\bar{c} &= \varphi, & s\varphi &= 0, \\ s\eta &= c, & sc &= 0. \\ s^2 &= 0, & sS &= 0. \end{aligned} \quad (7)$$

One sees that the field content of the model gives rise to a pair of BRST doublets[1]: $(\bar{c}, \varphi)$ and $(\eta, c)$. The whole set $(\bar{c}, \varphi, \eta, c)$ is called a BRST quartet. It should also be noted that the above action can be rewritten as a BRST exact term, that is,

$$S = \int d^2x \, s \left[ \partial_\mu \bar{c} \partial^\mu \eta - m^2 \bar{c} \eta \right]. \quad (9)$$

Here, we are adopting the Kugo-Ojima Hermiticity assignments [42–45]: $\varphi^\dagger = \varphi$, $\eta^\dagger = \eta$, $c^\dagger = c$, $\bar{c}^\dagger = -\bar{c}$, so as to have a Hermitian action, $S^\dagger = S$, considering the general rule for Hermitian conjugation $(AB)^\dagger = B^\dagger A^\dagger$ and the Grassmanian nature of the ghost fields. It is interesting to note that in the limit $m \to 0$, the model (6) becomes equivalent to the Maxwell action quantized in the Feynman gauge in $1 + 1$ dimensions.

---

[1] A BRST doublet is defined as a pair of fields $(\alpha, \beta)$ such that

$$s\alpha = \beta, \qquad s\beta = 0. \quad (8)$$

It can be shown that the fields of a BRST doublet do not contribute to the cohomology of the BRST operator [46].

From the action (6), we obtain the equations of motion:

$$\left(\Box + m^2\right)\Phi = 0, \quad \text{for } \Phi = \varphi, \eta, c, \bar{c}. \tag{10}$$

All fields obey the Klein-Gordon equation with the same mass $m$. We can perform the canonical quantization of these fields as follows:

$$\varphi(x) = \int \frac{dk}{\sqrt{(2\pi)2\omega_k}}\left(\varphi_k e^{-i(\omega_k t - kx)} + \varphi_k^\dagger e^{i(\omega_k t - kx)}\right), \tag{11}$$

where $\omega_k = k_0 = \sqrt{k^2 + m^2}$. Denoting the positive frequency modes in 1+1 Minkowski spacetime by

$$u_k = \frac{1}{\sqrt{2\pi\,2\omega_k}}e^{-i(\omega_k t - kx)}, \tag{12}$$

we can write the scalar field simply as

$$\varphi(x) = \int dk\left(\varphi_k u_k + \varphi_k^\dagger u_k^*\right). \tag{13}$$

It is helpful to introduce the Klein-Gordon inner product:

$$\langle f | g \rangle_{\mathrm{KG}} = i \int dx \left[f^*\left(\partial_t g\right) - \left(\partial_t f^*\right)g\right] \tag{14}$$

so that:

$$\begin{aligned}
\langle u_k | u_q \rangle_{\mathrm{KG}} &= \delta\left(k - q\right) \\
\langle u_k | u_q^* \rangle_{\mathrm{KG}} &= 0 \\
\langle u_k^* | u_q^* \rangle_{\mathrm{KG}} &= -\delta\left(k - q\right).
\end{aligned} \tag{15}$$

In the same vein, we can write a similar mode expansion for all the other fields:

$$\begin{aligned}
\eta(x) &= \int dk\left(\eta_k u_k + \eta_k^\dagger u_k^*\right), \\
c(x) &= \int dk\left(c_k u_k + c_k^\dagger u_k^*\right), \\
\bar{c}(x) &= \int dk\left(\bar{c}_k u_k - \bar{c}_k^\dagger u_k^*\right),
\end{aligned} \tag{16}$$

where we remark that the $\bar{c}_k^\dagger$ term has an opposite sign thanks to the difference in its Hermiticity assignment. The canonical commutation relations can be easily evaluated, being given by

$$\begin{aligned}
\left[\varphi\left(t, x\right), \partial^0 \eta\left(t, y\right)\right] &= i\delta\left(x - y\right), \\
\left[\eta\left(t, x\right), \partial^0 \varphi\left(t, y\right)\right] &= i\delta\left(x - y\right), \\
\left\{c\left(t, x\right), \partial^0 \bar{c}\left(t, y\right)\right\} &= i\delta\left(x - y\right), \\
\left\{\bar{c}\left(t, x\right), \partial^0 c\left(t, y\right)\right\} &= -i\delta\left(x - y\right),
\end{aligned} \tag{17}$$

where the minus sign in the last expression is due to the different sign obtained in the conjugate momentum $\pi_{\bar{c}} = \frac{\delta \mathcal{L}}{\delta(\partial_0 \bar{c})} = -\partial^0 c$. From the KG inner product, we have

$$\begin{aligned}
\varphi_k &= \langle u_k | \varphi \rangle_{\mathrm{KG}}, & \varphi_k^\dagger &= -\langle u_k^* | \varphi \rangle_{\mathrm{KG}}, \\
\eta_k &= \langle u_k | \eta \rangle_{\mathrm{KG}}, & \eta_k^\dagger &= -\langle u_k^* | \eta \rangle_{\mathrm{KG}},
\end{aligned}$$

$$\begin{aligned}
c_k &= \langle u_k | c \rangle_{\mathrm{KG}}, & c_k^\dagger &= -\langle u_k^* | c \rangle_{\mathrm{KG}}, \\
\bar{c}_k &= \langle u_k | \bar{c} \rangle_{\mathrm{KG}}, & \bar{c}_k^\dagger &= \langle u_k^* | \bar{c} \rangle_{\mathrm{KG}},
\end{aligned} \tag{18}$$

Using all these expressions, we can obtain the non-vanishing canonical commutation relations among the creation and annihilation operators:

$$\begin{aligned}
\left[\varphi_k, \eta_q^\dagger\right] &= \delta\left(k - q\right), & \left[\eta_k, \varphi_q^\dagger\right] &= \delta\left(k - q\right), \\
\left\{c_k, \bar{c}_q^\dagger\right\} &= -\delta\left(k - q\right), & \left\{\bar{c}_k, c_q^\dagger\right\} &= -\delta\left(k - q\right),
\end{aligned} \tag{19}$$

with all the remaining commutation relations vanishing. The Fock vacuum of Minkowski space $|0\rangle_M$ is defined as the state annihilated by all annihilation operators, i.e.,

$$\varphi_k |0\rangle_M = \eta_k |0\rangle_M = c_k |0\rangle_M = \bar{c}_k |0\rangle_M = 0. \tag{20}$$

Let us proceed with the characterization of the physical subspace $\mathcal{H}_{phys}$, Eq.(4). To that end we need to evaluate the BRST charge. From the Noether Theorem,

$$\left(\varphi(x)\frac{\delta}{\delta\bar{c}(x)} + c(x)\frac{\delta}{\delta\eta(x)}\right)S = \partial_\mu J_{BRST}^\mu(x), \tag{21}$$

for the BRST conserved current we get

$$J_{BRST}^\mu = \left(\partial^\mu c\right)\varphi - \left(\partial^\mu \varphi\right)c. \tag{22}$$

Defining the associated charge as the integral in space of the zeroth component of the conserved current, it follows

$$Q_{BRST} = \int dx\left[\left(\partial^0 c\right)\varphi - \left(\partial^0 \varphi\right)c\right]. \tag{23}$$

Making use of the mode expansion of these fields, one finds the expression for $Q_{BRST}$ in terms of creation and annihilation operators:

$$Q_{BRST} = -i\int dk\left(c_k \varphi_k^\dagger - \varphi_k c_k^\dagger\right). \tag{24}$$

From the above expression, one immediately obtains $Q_{BRST}^\dagger = Q_{BRST}$ and $Q_{BRST}^2 = 0$, two important properties of the BRST charge. Following Kugo-Ojima [42–45], we look at all commutation and anti-commutation relations of the creation and annihilation operators with the BRST charge, finding:

$$\begin{aligned}
\left[Q_{BRST}, \eta_k\right] &= ic_k, & \left\{Q_{BRST}, c_k\right\} &= 0, \\
\left[Q_{BRST}, \eta_k^\dagger\right] &= ic_k^\dagger, & \left\{Q_{BRST}, c_k^\dagger\right\} &= 0, \\
\left\{Q_{BRST}, \bar{c}_k\right\} &= -i\varphi_k, & \left[Q_{BRST}, \varphi_k\right] &= 0, \\
\left\{Q_{BRST}, \bar{c}_k^\dagger\right\} &= i\varphi_k^\dagger, & \left[Q_{BRST}, \varphi_k^\dagger\right] &= 0.
\end{aligned} \tag{25}$$

We see that $(\eta_k, c_k)$, $(\eta_k^\dagger, c_k^\dagger)$, $(\bar{c}_k, \varphi_k)$ and $(\bar{c}_k^\dagger, \varphi_k^\dagger)$ form BRST doublets. As such, they do not contribute to the cohomology of the BRST charge $Q_{BRST}$ [46]. Therefore, the unique state belonging to the physical subspace $\mathcal{H}_{phys}$ is the vacuum state $|0\rangle_M$, i.e., $\mathcal{H}_{phys} = \{|0\rangle_M\}$ [49].

We can thus appreciate the powerful and beautiful construction of Kugo-Ojima in order to deal with the BRST symmetry. We are ready now to face the more complex and non-trivial case of the Maxwell field.

## III. MAXWELL GAUGE THEORY FROM A BRST POINT OF VIEW

The main lines of the BRST approach were addressed in the last section, were we used the 1+1 model of Eq. (6) to discuss the BRST symmetry role in the physical states definition, via the Kugo-Ojima quartet mechanism. In this section, we move on to the Maxwell action describing photons in 1+3-dimensional Minkowski space endowed with metric $g_{\mu\nu} = \mathrm{diag}\,(+,-,-,-)$. In the sequel, we shall discuss the canonical quantization of the photon field, study its BRST symmetry, and investigate the possible violation of the Bell-CHSH inequality in a BRST invariant setup by considering the case of a squeezed light state.

### A. Quantization and BRST invariance

Let us consider the Maxwell action in 1+3-dimensional Minkowski spacetime

$$S_0 = \int d^4x \left(-\frac{1}{4} F_{\mu\nu} F^{\mu\nu}\right), \qquad (26)$$

where $A_\mu$ is the photon field, and its field strength tensor is given by $F_{\mu\nu} = \partial_\mu A_\nu - \partial_\nu A_\mu$. The above action is invariant under local Abelian gauge transformations given by $A'_\mu = A_\mu - \partial_\mu \omega$. In order to quantize the theory, we need to fix the gauge. Implementing the Faddeev-Popov procedure in the covariant Feynman gauge, one introduces the Nakanishi-Lautrup field $b$, obtaining the following gauge-fixed action:

$$S = \int d^4x \left(-\frac{1}{4} F_{\mu\nu} F^{\mu\nu} + b\,(\partial_\mu A^\mu) + \frac{b^2}{2} + \bar{c}\,\Box\,c\right), \quad (27)$$

where $c$ and $\bar{c}$ denote the Faddeev-Popov ghosts and we defined $\Box \equiv \partial_\mu \partial^\mu$. The gauge-fixed action (27) is invariant under the following BRST transformations

$$\begin{aligned} sA_\mu &= -\partial_\mu c, & sc &= 0, \\ s\bar{c} &= b, & sb &= 0, \\ s^2 &= 0, & sS &= 0. \end{aligned} \qquad (28)$$

As done in the previous section, we adopt the Kugo-Ojima Hermiticity assignments $A_\mu^\dagger = A_\mu$, $b^\dagger = b$, $c^\dagger = c$, $\bar{c}^\dagger = -\bar{c}$, so that $S^\dagger = S$. For the equations of motion:

$$\begin{aligned} \Box A_\mu &= \Box c = \Box \bar{c} = 0, \\ b + \partial_\mu A^\mu &= 0. \end{aligned} \qquad (29)$$

We can see that the Nakanishi-Lautrup field $b$ is an auxiliary field, enforcing the gauge-fixing condition, while all other fields satisfy the massless Klein-Gordon equation.

In the 1+3-dimensional Minkowski spacetime, we can define objects completely analogous to the ones defined in the previous section, in order to simplify the notation.

For instance, the Klein-Gordon inner product in 1+3-dimensional Minkowski spacetime is given by

$$\langle f|g\rangle_{\mathrm{KG}} = i \int d^3x \left[f^*\,(\partial_t g) - (\partial_t f^*)\,g\right], \qquad (30)$$

and the positive-frequency modes can be written as

$$u_k = \frac{1}{\sqrt{(2\pi)^3\,2\omega_k}} e^{-i\left(\omega_k t - \vec{k}\cdot\vec{x}\right)}. \qquad (31)$$

These modes satisfy the relations

$$\begin{aligned} \langle u_k|u_q\rangle_{\mathrm{KG}} &= \delta^3\left(\vec{k}-\vec{q}\right), \\ \langle u_k|u_q^*\rangle_{\mathrm{KG}} &= 0, \\ \langle u_k^*|u_q^*\rangle_{\mathrm{KG}} &= -\delta^3\left(\vec{k}-\vec{q}\right). \end{aligned} \qquad (32)$$

Let us proceed with the canonical quantization of the fields. First, we consider the gauge field mode expansion

$$A_i(x) = \int d^3k \sum_{\lambda=1}^{3} \epsilon_i^{(\lambda)}(k)\left(a_{\lambda k} u_k + a_{\lambda k}^\dagger u_k^*\right), \qquad (33)$$

$$A_0(x) = \int d^3k \left(a_{0k} u_k + a_{0k}^\dagger u_k^*\right), \qquad (34)$$

where $\omega_k = k_0 = |\vec{k}|$, since we are dealing now with massless fields. In the above expression, $\epsilon_i^{(\lambda)}(k)$ are polarization vectors. Here, $\epsilon_i^{(1)}(k)$ and $\epsilon_i^{(2)}(k)$ represent the transverse polarizations such that $\vec{\epsilon}^\alpha(k)\cdot\vec{\epsilon}^\beta(k) = \delta_{\alpha\beta}$ and $\vec{k}\cdot\vec{\epsilon}^\alpha(k) = 0$ for $\alpha,\beta = 1,2$, and $\epsilon_i^{(3)}(k) = k_i/|\vec{k}|$ is the longitudinal one. Concerning these polarization vectors, we can also write the following expression:

$$\sum_{\alpha=1,2} \epsilon_i^\alpha(k)\epsilon_j^\alpha(k) = \left(\delta_{ij} - \frac{k_i k_j}{|\vec{k}|^2}\right). \qquad (35)$$

Similarly, for the ghost fields we have,

$$c(x) = \int d^3k \left(c_k u_k + c_k^\dagger u_k^*\right), \qquad (36)$$

$$\bar{c}(x) = \int d^3k \left(\bar{c}_k u_k - \bar{c}_k^\dagger u_k^*\right), \qquad (37)$$

Remembering that $b = -\partial_\mu A^\mu$, it follows that the mode expansion of the Nakanishi-Lautrup field reads

$$b = i\int d^3k\,\omega_k\left(\left(a_{0k} u_k - a_{0k}^\dagger u_k^*\right) - \left(a_{3k} u_k - a_{3k}^\dagger u_k^*\right)\right). \qquad (38)$$

From the Lagrangian, Eq.(27), one obtains the following conjugate momenta: $\pi_i^A = -F_{0i}$, $\pi_0^A = b$, $\pi_c = \partial^0 \bar{c}$, $\pi_{\bar{c}} = -\partial^0 c$. Imposing the equal time canonical commutation relations:

$$\begin{aligned} \left[A_\mu(t,\vec{x}),\,\pi_\nu(t,\vec{y})\right] &= ig_{\mu\nu}\delta^3\,(\vec{x}-\vec{y}), \\ \{c(t,\vec{x}),\,\partial^0\bar{c}(t,\vec{y})\} &= i\delta^3\,(\vec{x}-\vec{y}), \\ \{\bar{c}(t,\vec{x}),\,\partial^0 c(t,\vec{y})\} &= -i\delta^3\,(\vec{x}-\vec{y}), \end{aligned} \qquad (39)$$

one gets

$$\left[a_{ik},\,a_{jq}^\dagger\right] = \delta_{ij}\delta^3\left(\vec{k}-\vec{q}\right), \quad \left[a_{0k},\,a_{0q}^\dagger\right] = -\delta^3\left(\vec{k}-\vec{q}\right),$$

$$\{c_k, \bar{c}_q^\dagger\} = -\delta^3\left(\vec{k} - \vec{q}\right), \qquad \{\bar{c}_k, c_q^\dagger\} = -\delta^3\left(\vec{k} - \vec{q}\right). \quad (40)$$

The minus signs in the commutation relations will give rise to unphysical states with negative norm [42–45]. In fact, consider the Fock vacuum of the Minkowski space $|0\rangle_M$, annihilated by all the annihilation operators:

$$a_{ik}|0\rangle_M = a_{0k}|0\rangle_M = c_k|0\rangle_M = \bar{c}_k|0\rangle_M = 0. \quad (41)$$

The whole Fock space is obtained by the action of the creation operators on the vacuum $|0\rangle_M$. This will include physical and unphysical states, and we need to employ the BRST charge in order to obtain the subspace of the physical states, as discussed before.

Following the same steps of the last section, we find the expression for the charge associated with the BRST symmetry:

$$Q_{BRST} = \int d^3k \left[c_k^\dagger\left(a_{0k} - a_{3k}\right) + c_k\left(a_{0k}^\dagger - a_{3k}^\dagger\right)\right]. \quad (42)$$

With the above expression, one can check that: $Q_{BRST}^\dagger = Q_{BRST}$, $Q_{BRST}^2 = 0$. The BRST charge, Eq.(42), also enjoys a remarkable property, concerning the unphysical modes [42–45]. Let $\mathcal{N}$ be the number operator counting all unphysical modes that are present in a state, that is,

$$\mathcal{N} = \int d^3k \left[a_{3k}^\dagger a_{3k} - a_{0k}^\dagger a_{0k} - c_k^\dagger \bar{c}_k - \bar{c}_k^\dagger c_k\right]. \quad (43)$$

It turns out that the number operator $\mathcal{N}$ can be written as the anti-commutator between the BRST charge and a certain operator $\mathcal{R}$:

$$\exists\, \mathcal{R} \quad \text{such that} \quad \mathcal{N} = \{Q_{BRST}, \mathcal{R}\}. \quad (44)$$

Explicitly, we have

$$\mathcal{R} = \int d^3k \left[\left(a_{0k}^\dagger + a_{3k}^\dagger\right)\bar{c}_k + \bar{c}_k^\dagger\left(a_{0k} + a_{3k}\right)\right]. \quad (45)$$

The physical subspace is defined by the cohomology of the BRST operator. Considering the property presented above, one can prove that the physical subspace $\mathcal{H}_{phys}$ cannot contain unphysical modes, being spanned by the positive norm states corresponding to the transverse polarizations $\epsilon_i^{(1)}$ and $\epsilon_i^{(2)}$ of the photon field.

In fact, suppose that the state $|\alpha\rangle$ is annihilated by the BRST charge, and that it contains some unphysical modes:

$$Q_{BRST}|\alpha\rangle = 0, \quad \mathcal{N}|\alpha\rangle = n|\alpha\rangle, \quad n \neq 0. \quad (46)$$

Therefore, from the previous property, we find:

$$n|\alpha\rangle = \mathcal{N}|\alpha\rangle = \{Q_{BRST}, \mathcal{R}\}|\alpha\rangle = Q_{BRST}\left[\mathcal{R}|\alpha\rangle\right], \quad (47)$$

which shows that BRST invariant states containing unphysical modes cannot belong to the physical subspace. Indeed, if Eq. (46) is valid, Eq. (47) implies that $|\alpha\rangle = \frac{1}{n}Q_{BRST}\left[\mathcal{R}|\alpha\rangle\right]$, making it a trivial state in the BRST cohomology.

The BRST invariant states belonging to the physical subspace are those generated by the creation operators $a_1^\dagger$ and $a_2^\dagger$ corresponding to the physical polarizations of the photon, $\epsilon_i^{(1)}$ and $\epsilon_i^{(2)}$, that is,

$$|f\rangle \in \mathcal{H}_{phys} \implies |f\rangle = \left(a_{1k}^\dagger\right)^m \left(a_{2q}^\dagger\right)^n |0\rangle_M. \quad (48)$$

We emphasize that, by construction, physical states must be annihilated by the BRST charge ($Q_{BRST}|f\rangle = 0$) without being in its image ($|f\rangle \neq Q_{BRST}|\cdot\rangle$). This follows by noticing that the BRST charge of Eq.(42) does not contain $(a_1^\dagger, a_2^\dagger)$:

$$\left[Q_{BRST}, a_{1k}^\dagger\right] = \left[Q_{BRST}, a_{2k}^\dagger\right] = 0. \quad (49)$$

Let us conclude this section by reminding that, as they stand, the states (48) are not truly normalizable. As is well known, this a consequence of the fact that quantum fields are rather singular quantities, being needed to be treated as operator-valued distributions [52]. To that end, and in view of the next construction, it is helpful to introduce the smeared operators [52]

$$a_1(f) = \int \frac{d^3k}{\sqrt{(2\pi)^3 2\omega_k}} a_{1k}\tilde{f}(k),$$

$$a_2(g) = \int \frac{d^3k}{\sqrt{(2\pi)^3 2\omega_k}} a_{2k}\tilde{g}(k),$$

$$a_1^\dagger(f) = \int \frac{d^3k}{\sqrt{(2\pi)^3 2\omega_k}} a_{1k}^\dagger\tilde{f}^*(k),$$

$$a_2^\dagger(g) = \int \frac{d^3k}{\sqrt{(2\pi)^3 2\omega_k}} a_{2k}^\dagger\tilde{g}^*(k), \quad (50)$$

where

$$\tilde{f}(p) = \int d^4x\, e^{-ipx} f(x), \quad \tilde{g}(p) = \int d^4x\, e^{-ipx} g(x), \quad (51)$$

and it is implicitly understood that we are considering $k_0 = \omega_k$ under all the integral signs in Eqs. (50).

The test functions $(f(x), g(x))$ belong to $\mathcal{C}_0^\infty\left(\mathbb{R}^4\right)$, the space of smooth infinitely differentiable functions with compact support. Moreover, as required by relativistic causality, the supports of $(f(x), g(x))$ are space-like separated. The space of test functions $\mathcal{C}_0^\infty\left(\mathbb{R}^4\right)$ is the adequate space[2]. to be employed in the study of the Bell inequalities in QFT [16–18]. The test functions are also normalizable, something we will use later on.

From the commutation relations (40) one gets

$$\left[a_\alpha(f), a_\beta^\dagger(g)\right] = \delta_{\alpha\beta}\langle f|g\rangle, \qquad \alpha, \beta = 1, 2 \quad (52)$$

where $\langle f|g\rangle$ is the Lorentz invariant scalar product

$$\langle f|g\rangle = \int \frac{d^3k}{(2\pi)^3 2\omega_k}\tilde{f}(\omega_k, \vec{k})\tilde{g}^*(\omega_k, \vec{k}),$$

---

[2] It is well-known that the Fourier transform $\hat{h}(p)$ of a test function $h(x) \in \mathcal{C}_0^\infty(\mathbb{R}^4)$ is a rapidly decreasing function al large $p$.

$$= \int \frac{d^4k}{(2\pi)^3}\theta(k_0)\delta(k^2)\tilde{f}(k)\tilde{g}^*(k). \quad (53)$$

In particular,

$$\left[a_\alpha(f), a_\beta^\dagger(f)\right] = \delta_{\alpha\beta}\langle f|f\rangle = \delta_{\alpha\beta}\|f\|^2, \quad (54)$$

where $\|f\|^2$ is the norm of $f$. Moreover, notice that by a simple rescaling, the test functions $f(x)$ can be always chosen to be normalized to 1, i.e.

$$f(x) \to \frac{1}{\|f\|}f(x) \quad \text{so that} \quad \|f\| = 1, \quad (55)$$

giving

$$\left[a_\alpha(f), a_\beta^\dagger(f)\right] = \delta_{\alpha\beta}. \quad (56)$$

It can be easily checked that, when acting on the vacuum state $|0\rangle_M$, the smeared operators $a_\beta^\dagger(f)$ give rise to truly normalizable states. Of course, $(a_\alpha(f), a_\beta^\dagger(g))$ are still left invariant by the BRST charge, namely

$$[Q_{BRST}, a_\alpha(f)] = \left[Q_{BRST}, a_\beta^\dagger(g)\right] = 0. \quad (57)$$

## B. BRST invariant construction of the Bell-CHSH inequality. Squeezed photon states

Let us face now the construction of the BRST invariant formulation of the Bell-CHSH inequality which, as already mentioned, will be investigated by using as probing state the normalized two-mode squeezed state, defined as

$$|\eta\rangle \equiv Ne^{\eta a_1^\dagger(f) a_2^\dagger(g)}|0\rangle_M. \quad (58)$$

The most general two-mode squeezed state, in terms of an $Sp(4,\mathbb{R})$ algebra, can be found in Ref. [53]. The special case of Eq. (58) already satisfies our current needs.

From Eq.(57), it follows that $|\eta\rangle$ belongs to the physical subspace, namely,

$$Q_{BRST}|\eta\rangle = 0, \qquad |\eta\rangle \neq Q_{BRST}|\cdot\rangle. \quad (59)$$

By expanding the exponential in Eq.(58), we can rewrite

$$|\eta\rangle = N\sum_{n=0}^\infty \eta^n |n_f, n_g\rangle, \quad (60)$$

where we defined the normalized states

$$|n_f, n_g\rangle = \frac{\left[a_1^\dagger(f)\right]^n}{\sqrt{n!}}\frac{\left[a_2^\dagger(g)\right]^n}{\sqrt{n!}}|0\rangle_M, \quad (61)$$

with $\langle m_f, m_g|n_f, n_g\rangle = \delta_{mn}$. Notice that both creation operators $a_1^\dagger$ and $a_2^\dagger$ have the same power in Eq.(60). From the normalization condition of $|\eta\rangle$, i.e., $\langle\eta|\eta\rangle = 1$, we immediately find $N = \sqrt{1-\eta^2}$, where $\eta \in [0,1[$ to guarantee convergence and thus the normalizability.

Let us proceed with the introduction of the four operators $(A_i, B_k)$, with $i, k = 1, 2$, Eq.(2). To that end we follow the procedure outlined in [15, 54] and rewrite expression (60) by considering the even and odd contributions, i.e.:

$$|\eta\rangle = N\sum_{n=0}^\infty \left[\eta^{2n}|2n_f, 2n_g\rangle + \eta^{2n+1}|2n_f+1, 2n_g+1\rangle\right] \quad (62)$$

and defining the operators $(A_i, B_j)$, with $i, j = 1, 2$ as

$$A_i|2n_f, \cdot\rangle = e^{i\alpha_i}|2n_f+1, \cdot\rangle,$$
$$A_i|2n_f+1, \cdot\rangle = e^{-i\alpha_i}|2n_f, \cdot\rangle,$$
$$B_j|\cdot, 2n_g\rangle = e^{i\beta_j}|\cdot, 2n_g+1\rangle,$$
$$B_j|\cdot, 2n_g+1\rangle = e^{-i\beta_j}|\cdot, 2n_g\rangle. \quad (63)$$

where $(\alpha_1, \alpha_2, \beta_1, \beta_2)$ are arbitrary real parameters which can be chosen at the best convenience. These parameters play the same role of the four unit arbitrary vectors entering the original Bell-CHSH for spin 1/2 [1–3]. Notice that the operator $A_i$ acts only on the first entry of the state $|n_f, n_g\rangle$, while the operator $B_j$ only on the second. By their very definition, it is immediate to see that $A_i^2 = B_j^2 = 1$, and that we have $[A_i, B_j] = 0$. Furthermore, reminding that $|n_f, n_g\rangle$ belong to the cohomology of the BRST charge $Q_{BRST}$, it follows that equation (63) implies that $(A_i, B_j)$ are BRST invariant operators, i.e.,

$$[Q_{BRST}, A_i] = [Q_{BRST}, B_j] = 0. \quad (64)$$

In fact, one can immediately see that

$$[Q_{BRST}, A_i]|2n_f, \cdot\rangle = e^{i\alpha_i}Q_{BRST}|2n_f+1, \cdot\rangle = 0,$$
$$[Q_{BRST}, B_j]|\cdot, 2n_g\rangle = e^{i\beta_j}Q_{BRST}|\cdot, 2n_g+1\rangle = 0. \quad (65)$$

These operators can be used to emulate an SU(2) pseudo-spin algebra, see [55, 56] and references therein for more details on this, effectively bringing one as close as possible to the original Bell-CHSH setup, even for the bosonic degrees of freedom we are considering now.

Now, let us consider the BRST invariant Bell-CHSH operator defined here by

$$\mathcal{C}_{CHSH} \equiv (A_1 + A_2)B_1 + (A_1 - A_2)B_2 \quad (66)$$

The goal here is to evaluate the correlation $\langle\mathcal{C}_{CHSH}\rangle_\eta \equiv \langle\eta|\mathcal{C}_{CHSH}|\eta\rangle$. In order to accomplish this task and check out if there is violation, we need to compute:

$$B_1|\eta\rangle = N\sum_{n=0}^\infty \left[\eta^{2n}e^{i\beta_1}|2n_f, 2n_g+1\rangle + \eta^{2n+1}e^{-i\beta_1}|2n_f+1, 2n_g\rangle\right]. \quad (67)$$

Acting now with $(A_1 + A_2)$ we find

$$(A_1 + A_2)B_1|\eta\rangle =$$
$$N\sum_{n=0}^\infty \left[\eta^{2n}\left(e^{i\alpha_1} + e^{i\alpha_2}\right)e^{i\beta_1}|2n_f+1, 2n_g+1\rangle\right.$$

$$+ \eta^{2n+1} \left(e^{-i\alpha_1} + e^{-i\alpha_2}\right) e^{-i\beta_1} |2n_f, 2n_g\rangle\big]. \quad (68)$$

Analogously, for the action of $(A_1 - A_2) B_2$ on $|\eta\rangle$. Using the normalization $\langle m_f, m_g | n_f, n_g \rangle = \delta_{mn}$, we find

$$\langle \mathcal{C}_{CHSH} \rangle_\eta = N^2 \sum_{n=0}^{\infty} \eta^{4n+1} 2\,\Omega\left(\alpha_1, \alpha_2, \beta_1, \beta_2\right), \quad (69)$$

where we defined

$$\Omega\left(\alpha_1, \alpha_2, \beta_1, \beta_2\right) \equiv \cos(\beta_1 + \alpha_1) + \cos(\beta_1 + \alpha_2)$$
$$+ \cos(\beta_2 + \alpha_1) - \cos(\beta_2 - \alpha_2). \quad (70)$$

Therefore, we obtain the final result

$$\langle \mathcal{C}_{CHSH} \rangle_\eta = \frac{2\eta}{1+\eta^2}\,\Omega\left(\alpha_1, \alpha_2, \beta_1, \beta_2\right). \quad (71)$$

It remains now to choose the free parameters $(\alpha_i, \beta_j)$. Following [15, 54], we take $(\alpha_1, \alpha_2, \beta_1, \beta_2) = (0, +\pi/2, -\pi/4, +\pi/4)$, yielding the maximum value for $\Omega$, namely, $\Omega = 2\sqrt{2}$. Thus,

$$\langle \mathcal{C}_{CHSH} \rangle_\eta = 2\frac{2\sqrt{2}\,\eta}{1+\eta^2}. \quad (72)$$

There is a Bell-CHSH inequality violation for this choice of parameters $(\alpha_1, \alpha_2, \beta_1, \beta_2)$ whenever $\sqrt{2}-1 < \eta < 1$. As $\eta$ increases, the violation becomes higher, staying close to Tsirelson's bound $(2\sqrt{2})$ for $\eta \approx 1$. Notice also that the maximization procedure for the Bell-CHSH inequality translates into the necessary stationary condition

$$\frac{\partial}{\partial \eta} \langle \mathcal{C}_{CHSH} \rangle_\eta = 0, \quad (73)$$

which is a self-consistent gap equation for the squeezing parameter $\eta$, supplemented with $\frac{\partial^2}{\partial \eta^2} \langle \mathcal{C}_{CHSH} \rangle_\eta > 0$.

We see therefore that the squeezed state $|\eta\rangle$ might lead to a strong violation of the Bell-CHSH inequality even in the case of the free Maxwell theory, a fact already emphasized in the pioneering work [15–19]. Moreover, thanks to the Kugo-Ojima construction, everything has been realized in a manifestly BRST invariant way.

## IV. A HELPFUL COMPARISON WITH QUANTUM MECHANICS

For a better understanding of the result (72), it is helpful to provide a comparison with Quantum Mechanics. Consider an entangled state of two spin 1/2 particles,

$$|\psi\rangle = \left(\frac{|+\rangle_a |-\rangle_b - r|-\rangle_a |+\rangle_b}{\sqrt{1+r^2}}\right), \qquad \langle \psi | \psi \rangle = 1, \quad (74)$$

where $r$ stands for a real positive parameter, $r > 0$. We can introduce the operators $(A_i, B_k)$ in the same way as before, that is,

$$A_i|+\rangle_a = e^{i\alpha_i}|-\rangle_a, \qquad A_i|-\rangle_a = e^{-i\alpha_i}|+\rangle_a,$$

$$B_k|+\rangle_b = e^{-i\beta_k}|-\rangle_b, \qquad B_k|-\rangle_b = e^{i\beta_k}|+\rangle_b. \quad (75)$$

Then,

$$A_i B_k |\psi\rangle = \left(\frac{-re^{-i(\alpha_i + \beta_k)}|+\rangle_a |-\rangle_b + e^{i(\alpha_i + \beta_k)}|-\rangle_a |+\rangle_b}{\sqrt{1+r^2}}\right) \quad (76)$$

and

$$\langle \psi | A_i B_k | \psi \rangle = -2\frac{r}{1+r^2}\cos(\alpha_i + \beta_k). \quad (77)$$

Using the values $(\alpha_1 = 0, \alpha_2 = \frac{\pi}{2}, \beta_1 = -\frac{\pi}{4}, \beta_2 = \frac{\pi}{4})$, we get

$$|\langle \mathcal{C}_{CHSH} \rangle_\psi| = 2\frac{2\sqrt{2}\,r}{1+r^2}, \quad (78)$$

which is very similar to Eq.(72). The parameter $r$ plays now exactly the same role of the squeezing parameter $\eta$. There is violation whenever we have

$$\sqrt{2}-1 < r < \sqrt{2}+1. \quad (79)$$

The maximum violation is attained when $r = 1$, $|\langle \mathcal{C}_{CHSH} \rangle_\psi| = 2\sqrt{2}$, in which case the state $|\psi\rangle$ becomes maximally entangled, coinciding in fact with the Bell singlet state.

In much the same way, tracing out one photon polarization, it is quickly checked that the reduced density matrix $\rho_1$ obtained from $\rho_{12} = |\eta\rangle\langle\eta|$, i.e.,

$$\rho_1 = \mathrm{Tr}_2(\rho_{12}) = (1-\eta^2) \sum_{n=0}^{\infty} \eta^{2n} |n_f\rangle\langle n_f|, \quad (80)$$

gives

$$\rho_1^2 = (1-\eta^2)^2 \sum_{n=0}^{\infty} \eta^{4n} |n_f\rangle\langle n_f|. \quad (81)$$

Therefore,

$$\mathrm{Tr}\rho_1^2 = \frac{1-\eta^2}{1+\eta^2}, \quad (82)$$

from which one can immediately see that $\rho_1$ becomes very impure when $\eta \approx 1$. This simple analogy gives a rather nice understanding of the role played by the parameter $\eta$. As $\eta$ increases, the squeezed state $|\eta\rangle$ becomes more and more entangled, resulting in a strong violation of the Bell-CHSH inequality when $\eta \approx 1$.

One can also expect this pattern of violation of the Bell-CHSH inequality considering the entanglement entropy. The reduced density matrix (80) yields

$$S = -\mathrm{Tr}\rho_1 \ln \rho_1 = -\ln\left(1-\eta^2\right) - \frac{\eta^2 \ln \eta^2}{1-\eta^2}. \quad (83)$$

This entropy is a monotonically increasing function of $\eta$ and diverges for $\eta \to 1$, confirming that the system is highly entangled in this limit.

## V. EXTENDING THE BRST INVARIANT SETUP TO THE ABELIAN HIGGS MODEL

We extend now the results obtained in the previous sections to the Abelian Higgs model. In this first paper, we restrict ourselves to the quadratic approximation which, according to the general results [15–19], already displays all features enabling for the violation of the Bell-CHSH inequality. Said otherwise, we omit interactions for now. For the starting action we write

$$\mathcal{L}_{quad} = -\frac{1}{4}F_{\mu\nu}F^{\mu\nu} + \frac{m^2}{2}A_\mu A^\mu + \frac{1}{2}\left(\partial_\mu\rho\right)^2 + mA_\mu\partial^\mu\rho$$
$$+ \frac{1}{2}\left(\partial_\mu H\right)^2 - \frac{m_H^2}{2}H^2 + \frac{b^2}{2} + b\left(\partial_\mu A^\mu - m\rho\right)$$
$$- \partial_\mu\bar{c}\,\partial^\mu c + m^2\bar{c}c. \tag{84}$$

with

$$S_{quad} = \int d^4x\mathcal{L}_{quad}, \tag{85}$$

where $A_\mu$ is the Abelian gauge field, $H$ the Higgs scalar field, $\rho$ the Goldstone boson and $(\bar{c}, c)$ the Faddeev-Popov ghosts. Expression (84) can be obtained by expanding the $U(1)$ Higgs model

$$\mathcal{L} = -\frac{1}{4}F_{\mu\nu}F^{\mu\nu} + |(\partial_\mu\phi + ieA_\mu\phi)|^2 - \frac{\lambda}{2}\left(|\phi|^2 - \frac{v^2}{2}\right)^2 + \mathcal{L}_{FP} \tag{86}$$

around the vacuum configuration

$$\phi(x) = \frac{1}{\sqrt{2}}\left(v + H + i\rho\right) \tag{87}$$

while retaining the quadratic terms. The gauge boson mass and the Higgs mass are given, respectively, by

$$m^2 = e^2v^2, \qquad m_H^2 = \lambda v^2. \tag{88}$$

In the above expressions, $e$ is the electric charge and $\lambda > 0$ the self-quartic scalar coupling, while $v/\sqrt{2}$ is the minimum of the Higgs potential. Moreover, $\mathcal{L}_{FP}$ stands for the Faddeev-Popov gauge-fixing in 't Hooft gauge, i.e.,

$$\mathcal{L}_{FP} = \xi\frac{b^2}{2} + b\left(\partial_\mu A^\mu - \xi ev\rho\right) + \bar{c}\left(\partial_\mu\partial^\mu + \xi e^2v^2\right)c. \tag{89}$$

In the following, the value $\xi = 1$ will be adopted. As usual, $b$ is the Nakanishi-Lautrup auxiliary field.

As shown by Kugo-Ojima [42–45], the quadratic action $S_{quad}$, (85), is perfectly suited for the analysis of the BRST cohomology. In fact, it follows that $S_{quad}$ is left invariant by the nilpotent BRST transformations[3]

$$sA_\mu = -\partial_\mu c, \quad s\bar{c} = b, \quad s\rho = mc,$$

---

[3] As one can easily figure out, the transformations (90) are nothing but the linearized version of the BRST transformations which leave invariant the full action (86) [42–45].

$$sH = 0, \qquad sb = 0, \quad sc = 0,$$
$$s^2 = 0, \quad sS_{quad} = 0. \tag{90}$$

Again, we adopt the Kugo-Ojima Hermiticity assignments: $A_\mu^\dagger = A_\mu$, $c^\dagger = c$, $\bar{c}^\dagger = -\bar{c}$, $H^\dagger = H$, $\rho^\dagger = \rho$, $b^\dagger = b$, so that $S^\dagger = S$. The equations of motion here are given by:

$$\left(\Box + m^2\right)\Phi = 0,$$
$$\left(\Box + m_H^2\right)H = 0,$$
$$b + \partial_\mu A^\mu - m\rho = 0, \tag{91}$$

where $\Phi = A_\mu, \rho, c, \bar{c}$. Besides the Nakanishi-Lautrup field $b$, we see that the Higgs field obeys the Klein-Gordon equation with mass $m_H$, while the other fields satisfy the Klein-Gordon equations with the same mass $m$.

As pointed out by Kugo-Ojima [42–45], the quadratic action $S_{quad}$ (85) is diagonalizable by introducing the following field combination:

$$U_\mu \equiv A_\mu - \frac{1}{m^2}\partial_\mu b + \frac{1}{m}\partial_\mu\rho. \tag{92}$$

The field $U_\mu$ has very interesting features. First of all, it is BRST invariant, $sU_\mu = 0$. Looking at the equations of motion, we see that it satisfies the Klein-Gordon equation with mass $m$, $\left(\Box + m^2\right)U_\mu = 0$ as well as the transversality condition, given by $\partial_\mu U^\mu = 0$. Therefore, the vector field $U_\mu$ is a BRST invariant Proca field with mass $m$. It identifies the three BRST invariant degrees of freedom of the gauge vector boson, dynamically massive by means of the Higgs mechanism.

When rewritten in terms of the new vector field $U_\mu$, the action takes the following nice form:

$$\mathcal{S}_{quad} = \mathcal{S}_{phys} + \mathcal{S}_{unphys}, \tag{93}$$

where we defined the physical part, $\mathcal{S}_{phys}$, as that being given by the BRST invariant fields

$$sH = sU_\mu = 0, \tag{94}$$

namely,

$$\mathcal{S}_{phys} = \int d^4x\left[-\frac{1}{4}\mathcal{F}_{\mu\nu}(U)\mathcal{F}^{\mu\nu}(U) + \frac{m^2}{2}U_\mu U^\mu\right.$$
$$\left.+ \frac{1}{2}\left(\partial_\mu H\right)^2 - \frac{m_H^2}{2}H^2\right], \tag{95}$$

where the field strength now is $\mathcal{F}_{\mu\nu}(U) = \partial_\mu U_\nu - \partial_\nu U_\mu$.

The remaining terms are collected in $\mathcal{S}_{unphys}$, that can be written as

$$\mathcal{S}_{unphys} = \int d^4x\left[\frac{1}{2m^2}b\left(\Box + m^2\right)b - \frac{b}{m}\left(\Box + m^2\right)\rho\right.$$
$$\left.+ \bar{c}\left(\Box + m^2\right)c\right]. \tag{96}$$

Notice that this action is BRST exact, i.e., it can be rewritten as:

$$\mathcal{S}_{unphys} = \int d^4x \ s\left(\frac{1}{m}\partial_\mu\bar{c}\,\partial^\mu\rho - m\bar{c}\rho + \frac{\bar{c}b}{2} + \frac{1}{2m^2}\bar{c}\Box b\right) \tag{97}$$

Remarkably, the fields entering the above expression give rise to a BRST quartet, *i.e.*

$$s\bar{c} = b, \quad s\rho = mc, \quad sb = 0, \quad sc = 0. \tag{98}$$

As such, they do not contribute to the physical subspace, spanned by $(H, U_\mu)$. This statement expresses the content of the Kugo-Ojima analysis of the Higgs model [42–45].

By the same Noether procedure already adopted in the last sections, we can evaluate the charge associated with the BRST symmetry (90), obtaining:

$$Q_{BRST} = \int d^3x \left[ \left( \partial^0 c \right) b - \left( \partial^0 b \right) c \right]. \tag{99}$$

The physical subspace is constructed by acting on the vacuum state with the creation operators of the mode expansion of $(H, U_\mu)$:

$$|\text{phys}\rangle = \left( h^\dagger \right)^n \left( a_\lambda^\dagger \right)^p |0\rangle_M, \tag{100}$$

where $Q_{BRST}|\text{phys}\rangle = 0$, $|\text{phys}\rangle \neq Q_{BRST}|\cdot\rangle$. As before, $|0\rangle_M$ stands for the Fock vacuum of the Minkowski space, being annihilated by all annihilation operators. For the mode expansion of the fields $(H, U_\mu)$ we have

$$H(x) = \int \frac{d^3\vec{k}}{\sqrt{(2\pi)^3 2\omega_k^H}} \left( h_k e^{-ikx} + h_k^\dagger e^{+ikx} \right), \tag{101}$$

and

$$U_\mu(x) = \int \frac{d^3\vec{k}}{\sqrt{(2\pi)^3 2\omega_k^m}} \sum_{\lambda=1}^{3} \epsilon_\mu^{(\lambda)}(k) \left( a_{\lambda k} e^{-ikx} + a_{\lambda k}^\dagger e^{+ikx} \right), \tag{102}$$

with $\omega_k^H = \sqrt{|\vec{k}|^2 + m_H^2}$ and $\omega_k^m = \sqrt{|\vec{k}|^2 + m^2}$. The creation and annihilation operators satisfy the commutation relations $\left[ h_k, h_q^\dagger \right] = \delta^{(3)} \left( \vec{k} - \vec{q} \right)$ and $\left[ a_{\lambda k}, a_{\rho q}^\dagger \right] = \delta_{\lambda\rho} \delta^{(3)} \left( \vec{k} - \vec{q} \right)$. The three polarization vectors $\epsilon_\mu^{(\lambda)}(k)$ obey the relations [57] $k^\mu \epsilon_\mu^{(\lambda)}(k) = 0$ for $\lambda = 1, 2, 3$ as well as $\sum_{\lambda=1}^{3} \epsilon_\mu^{(\lambda)}(k) \epsilon_\nu^{(\lambda)}(k) = (g_{\mu\nu} - \frac{k_\mu k_\nu}{m^2})$.

### A. Construction of the BRST invariant squeezed states and violation of the Bell-CHSH inequality in the Abelian Higgs model

In order to construct normalizable squeezed states, we proceed as in the case of the Maxwell theory and introduce the smeared operators

$$h(g) = \int \frac{d^3\vec{k}}{\sqrt{(2\pi)^3 2\omega_k^H}} \tilde{g}(k) h_k,$$

$$a_1(f_1) = \int \frac{d^3\vec{k}}{\sqrt{(2\pi)^3 2\omega_k^m}} \tilde{f}_1(k) a_{1k},$$

$$a_2(f_2) = \int \frac{d^3\vec{k}}{\sqrt{(2\pi)^3 2\omega_k^m}} \tilde{f}_2(k) a_{2k},$$

$$a_3(f_3) = \int \frac{d^3\vec{k}}{\sqrt{(2\pi)^3 2\omega_k^m}} \tilde{f}_3(k) a_{3k}, \tag{103}$$

as well as their respective Hermitian conjugates. As before, the test functions $(g, f_i)$ belong to the space $\mathcal{C}_0^\infty(\mathbb{R}^4)$. In terms of the smeared operators (103), the canonical commutation relations become

$$\left[ h(g), h^\dagger(g') \right] = \langle g|g' \rangle,$$
$$\left[ a_\lambda(f), a_{\lambda'}^\dagger(f') \right] = \delta_{\lambda\lambda'} \langle f|f' \rangle, \tag{104}$$

where $\langle g|g' \rangle, \langle f|f' \rangle$ stand for the Lorentz invariant inner products defined in Eq. (53).

In the present case we can construct several types of squeezed states as, for example,

$$|\xi\rangle \equiv (1 - \xi^2)^{1/2} e^{\xi h^\dagger(f) a_3^\dagger(g)} |0\rangle_M,$$
$$|\zeta\rangle \equiv (1 - \zeta^2)^{1/2} e^{\zeta a_1^\dagger(f) a_2^\dagger(g)} |0\rangle_M. \tag{105}$$

where $(\xi, \zeta) \in [0, 1[$ to guarantee convergence.

To introduce the four operators $(A_i, B_j)$, $i, j = 1, 2$, Eq.(2), we repeat the same steps done in the Maxwell case. For example, considering the state $|\xi\rangle$, we write

$$|\xi\rangle = N_\xi \sum_{n=0}^{\infty} \left[ \xi^{2n} |2n_f, 2n_g\rangle + \xi^{2n+1} |2n_f + 1, 2n_g + 1\rangle \right] \tag{106}$$

where $N_\xi = (1 - \xi^2)^{1/2}$, and $|n_f, n_g\rangle = \frac{(h^\dagger(f) a_3^\dagger(g))^n}{n!} |0\rangle_M$. Let us consider the same Bell setup as before (63), *i.e.*,

$$A_i|2n_f, \cdot\rangle = e^{i\alpha_i}|2n_f + 1, \cdot\rangle, \quad A_i|2n_f + 1, \cdot\rangle = e^{-i\alpha_i}|2n_f, \cdot\rangle,$$
$$B_j|\cdot, 2n_g\rangle = e^{i\beta_j}|\cdot, 2n_g + 1\rangle, \quad B_j|\cdot, 2n_g + 1\rangle = e^{-i\beta_j}|\cdot, 2n_g\rangle,$$

where $(\alpha_1, \alpha_2, \beta_1, \beta_2)$ are arbitrary real parameters. By construction, these operators fulfill $A_i^2 = B_j^2 = 1$, $[A_i, B_j] = 0$ and are BRST invariant, $[Q_{BRST}, A_i] = [Q_{BRST}, B_j] = 0$. For the Bell-CHSH inequality violation, we now get

$$\langle \mathcal{C}_{CHSH} \rangle_\xi = 2\frac{2\sqrt{2}\,\xi}{1 + \xi^2}, \qquad \langle \mathcal{C}_{CHSH} \rangle_\zeta = 2\frac{2\sqrt{2}\,\zeta}{1 + \zeta^2} \tag{107}$$

attaining the maximum value for $(\xi, \zeta) \approx 1$. Obviously, the same remarks about the entanglement entropy that were made before for the photon case can be done here.

### VI. CONCLUSIONS AND FUTURE PERSPECTIVES

In this work the Bell-CHSH inequality has been investigated within the context of gauge field theories in Fock space. Relying on the construction outlined by Kugo-Ojima [42–45], we have been able to cast the Bell-CHSH inequality in a manifestly BRST invariant way.

According to eqs. (4), (5), this has required the use of the BRST invariant physical subspace, identified through

the cohomology of the BRST charge, $Q_{BRST}$, in Fock space as well as the BRST invariant construction of the four operators $(A_i, B_k)$ entering the Bell-CHSH correlator, Eq. (1). The examples of the Maxwell gauge field and of the Abelian Higgs model have been scrutinized. The violation of the Bell-CHSH inequality has been investigated by taking as probing states the BRST invariant squeezed states built with the creation operators corresponding to the physical polarizations of the photon and of the massive vector boson of the Higgs model.

In general, it seems fair to state that the study of the Bell-CHSH inequality in relativistic Quantum Field Theory is still at its infancy. It represents a great challenge, from theoretical, phenomenological and experimental points of view. Till now, the inequality has been investigated by considering free theories which, as proven by [15–19], already exhibit violation. Needless to say, the next step is to take interactions into account and find out how the violation depends on the coupling constants. Let us mention here that the recent formulation of the Bell-CHSH inequality within the Feynman path integral [21] might open the possibility of studying the inequality through the dictionary of Feynman diagrams. This might lead to an extension of the present analysis to non-Abelian gauge theories as well. Furthermore, a general setup to investigate the Bell-CHSH inequality violation for the vacuum state in the realm of QFT was proposed recently [64], opening a new path to investigate Bell-CHSH inequalities in a more general setting, which can also include the interacting case.

One may envisage that including quantum corrections will lead to parameters running according to the energy scale. Thinking about a renormalization group (RG) equation for the Bell correlators, taking into account the fact that these operators dimensionless and square to 1, it is very reasonable to assume that the Bell operators $A_i$, $B_j$ will not get anomalous dimensions. However, since we are evaluating these correlators not in the vacuum but in a squeezed state, the couplings defining that state could run according to an RG equation, next to the standard couplings of the underlying QFT.

To that end, it might be interesting to observe that one can actually rewrite Eq. (58) (see e.g. [58]) as

$$|\sigma\rangle = \mathcal{K}_\sigma |0\rangle_M, \qquad (108)$$

where $\sigma \in [0, 1[$, and $\mathcal{K}_\sigma$ is the unitary operator

$$\mathcal{K}_\sigma = \frac{1}{\cosh \sigma} \exp\left[\frac{\sigma}{2}(a_1^\dagger(f)a_2^\dagger(g) - a_1(f)a_2(g))\right], \quad (109)$$

upon identifying $\zeta = \tanh \sigma$. The Bell-CHSH expression can then be recast in the form

$$\langle\sigma|A_i B_j|\sigma\rangle = {}_M\langle 0|\mathcal{A}_i^K \mathcal{B}_j^K|0\rangle_M, \qquad (110)$$

where $\mathcal{A}_i^K = \mathcal{K}_\sigma^\dagger A_i \mathcal{K}_\sigma$ and $\mathcal{B}_j^K = \mathcal{K}_\sigma^\dagger B_j \mathcal{K}_\sigma$ can be thought as a kind of dressed Bell operators, now depending on the squeezing parameter. Eq. (110) can be interpreted, in a suggestive way, as the two-point correlation function in the vacuum of the composite dressed operators $\mathcal{A}_i^K$ and $\mathcal{B}_j^K$, which are unitarily equivalent to the original Bell operators.

We point out there that the squeezing parameter is a measurable quantity, see e.g. [59] and references therein for recent experimental progress regarding this hard to control parameter. Since these squeezing parameters enter the amount of violation, this would also mean the saturation of Tsirelson's bound found for the free theory might be corrected again upon including radiative corrections.

One challenge will be to construct suitable dichotomic Bell-CHSH operators like the used $A_i$, $B_j$ for a generic quantum field theory, let stand alone a gauge theory. One could proceed along the lines of [56, 60] to construct suitable pseudo-spin Bell operators and investigate to what extent some version of such operators can be put under the path integral sign. In order to make the transition to a path integral language, it might also be fruitful to express the squeezed state in terms of field and conjugate momentum operators, following [61], after which a suitable perturbation theory could be developed, following e.g. [62, 63]. This will be the subject of future work.

### ACKNOWLEDGMENTS

The authors would like to thank the Brazilian agencies CNPq and FAPERJ for financial support. S.P. Sorella is a level 1 CNPq researcher under the contract 301030/2019-7. M.S. Guimaraes is a level 2 CNPq researcher under the contract 310049/2020-2. G. Peruzzo is a FAPERJ postdoctoral fellow in the *Pós-Doutorado Nota 10* program under the contracts E-26/205.924/2022 and E-26/205.925/2022.

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
