# Peer review of "BRST invariant formulation of the Bell-CHSH inequality in gauge field theories"

_SciPost Physics_

## Round 1 · Referee Report · Reinhard Alkofer (Referee 1) · 2023-7-17

Strengths

1- The authors discuss very nicely how the quantum entanglement of squeezed states in quantum gauge field theories can be reformulated to be based on BRST-invariant states. The manuscript is very well written, and all the main arguments can be followed step by step.

Weaknesses

1- The remarks on the entanglement properties of the Minkowski vacuum when expressed by means of the Rindler wedges should either be elaborated more or left out. (This remark does not apply if in the meantime ref. [52] has been published.

Report

As argued above this is a well written manuscript on an important issue in quantum field theory.
I recommend to accept this article for publication.

Requested changes

1- A minor remark (optional change): In the introduction in the paragraph on proposed experimental tests the authors might wish to cite Bertlmann:2001sk, i.e.,
R.~A.~Bertlmann and B.~C.~Hiesmayr,
Bell inequalities for entangled kaons and their unitary time evolution,''
Phys. Rev. A \textbf{63} (2001), 062112
doi:10.1103/PhysRevA.63.062112
[arXiv:hep-ph/0101356 [hep-ph]].
as a quite early example.

---

## Round 1 · Referee Report · Anonymous (Referee 2) · 2023-8-11

Strengths

1 - clear and explicit presentation of the construction of BRST-invariant formulations of the Bell-CHSH inequality for two examples of non-interacting Abelian gauge theories
2 - nice interpretation of the presented results by drawing a comparison with a simple quantum mechanical example

Weaknesses

1 - considerable redundancies in the constructions in chapter II, III and V

Report

I recommend the submitted manuscript for publication in SciPost Phys. I is well-written and allows the reader the follow all steps of the explicit computation.

Requested changes

I consider below requested changes to be optional:

The authors might want to consider removing some of the mentioned redundancies which are rooted in the analogous constructions of secs. II, III and IV. This would increase the readability of the manuscript, since in the current version, one might get the feeling that one is reading the same thing over and over again. Due to the explicitness of the computations in secs. III and V, in my opinion the example given in sec. II does not add a lot to the manuscript, and might even be dropped.

Furthermore, I have some additional minor remarks:
- the authors might want to be a bit more careful with distinguishing between three- and four-vectors in eqs. (11)-(14)
- it is unclear to me how the authors arrive at the conclusion directly below eq. (47) from the respective eqs. above. maybe they can elaborate their argument a bit further.

---

## Editorial Decision

resubmitted